

# Distribution and evolution of the western European water frogs (genus *Pelophylax*) from Catalonia, northeastern Spain

Bernat Burriel-Carranza[1,2], Carolina Molina-Duran[2], Karin Tamar[3], Laia Pérez-Sorribes[4], Jhulyana López-Caro[5], Mar Cirac[2], Daniel Fernandez-Guiberteau[5] and Salvador Carranza[2]

[1] Museu de Ciències Naturals de Barcelona, Barcelona, Catalonia, Spain
[2] Institute of Evolutionary Biology (CSIC-Universitat Pompeu Fabra), Barcelona, Catalonia, Spain
[3] The Steinhardt Museum of Natural History, Israel National Center for Biodiversity Studies, Tel Aviv University, Tel Aviv, Israel
[4] Estación Biológica de Doñana (CSIC), Sevilla, Andalucia, Spain
[5] Centre de Recerca i Educació Ambiental de Calafell (CREAC-GRENP-Ajuntament de Calafell), Tarragona, Catalonia, Spain

Corresponding authors
Bernat Burriel-Carranza,
bernatburriel@gmail.com
Salvador Carranza,
salvador.carranza@ibe.upf-csic.es

## ABSTRACT

European water frogs from the genus *Pelophylax* are particular among amphibians as they can produce hybrids (named kleptons) reproducing by hybridogenesis. Four klepton species have been described in Europe: *Pelophylax* kl. *esculentus*, *P.* kl. *hispanicus*, *P.* kl. *grafi*, and the putative klepton PK. While most of these kleptons originated naturally in areas where the parental species occur in sympatry, human-mediated translocations of water frogs across Europe have altered this dynamic. As a result, several *Pelophylax* species and kleptons are now found outside their natural ranges, posing a threat to autochthonous water frogs. Additionally, the subtle morphological differences between *Pelophylax* species make hybrid identification and, thus, conservation difficult. In the present study, we analyzed 423 specimens from 54 populations sampled across Catalonia and implemented a two-step molecular method to identify all species of water frogs present in Catalonia. We also examined the mitochondrial genome of the hybrid *Pelophylax* kl. *grafi* to obtain new insights into their reproductive system and spatial structure. Despite the large number of samples analyzed, only the native *P. perezi* and its klepton *P.* kl. *grafi* were found, with the proportion of the latter being unexpectedly high. Results showed a high misidentification rate based on morphology compared to molecular methods, indicating that identification of *P.* kl. *grafi* through morphological characters is unreliable. Furthermore, the mitochondrial DNA of hybrid specimens entirely belonged to *P. perezi* and showed high intra-specific variability. This suggests either a single hybridization event involving a male *P. ridibundus* or *P.* kl. *esculentus* and a female *P. perezi,* or that *P. ridibundus* mitochondrial DNA has been eliminated from the *P.* kl. *grafi* germline by adaptive or non-adaptive processes. This study offers new insights into the distribution and composition of the North Iberian *Pelophylax* hybridogenetic complex, providing comprehensive sampling across one of the main entry points of the complex into the Iberian Peninsula.

## INTRODUCTION

The *Pelophylax* genus, widely distributed across the Western Palearctic, is known for its complex reproductive systems, particularly in contact zones where hybridogenetic mechanisms occur (*Berger, 1988*; *Berger, 1967*; *Dubois & Günther, 1982*; *Graf & Polls-Pelaz, 1989*; *Holsbeek & Jooris, 2010*; *Schultz, 1969*). In these systems, hybrids often rely on individuals from one of the parental species for reproduction, acting as sexual parasites and earning the designation "kleptons" (*Berger, 1988*). Hybridogenetic reproduction involves the production of hemiclonal offspring, a process where one parental genome is inherited clonally while the other is replaced by a genome from a sympatric parental species (*Dudzik et al., 2023*; *Graf & Polls-Pelaz, 1989*; *Ogielska, 1994*; *Pagano, Joly & Hotz, 1997*; *Tunner & Heppich, 1981*; *Tunner & Heppich-Tunner, 1991*). The absence of recombination (*Lehtonen et al., 2013*) means that a "fixed-in-time" genome is transmitted from one generation to another (*i.e.,* although accumulating mutations), with records of having persisted for at least 25,000 generations (*Guex, Hotz & Semlitsch, 2002*). This unique mode of reproduction, which merges aspects of both sexual and asexual reproduction, allows hybrid lineages to persist across generations.

The hybridogenetic reproductive systems display considerable geographic variation in how parental genomes are excluded, which directly influences hybrid population structure (*Plötner et al., 2008*; *Skierska et al., 2023*). For example, the L-E system (crosses between *P. lessonae* and *P.* kl. *esculentus*) predominantly excludes the *P. lessonae* genome and retains the parental *P. ridibundus* genetic material, whereas in the R-E system (*P. ridibundus* x *P.* kl. *esculentus*), the elimination of the *P. ridibundus* genome occurs less efficiently, leading to variable hybrid and parental offspring proportions (*Berger, 1973*; *Dubey & Dufresnes, 2017*; *Graf, Karch & Moreillon, 1977*; *Plötner et al., 2008*). Such reproductive flexibility highlights the adaptive potential of kleptons, which are capable of evolving diverse strategies depending on the local parental species or even persisting without them in full-klepton populations (*Christiansen, 2009*) (Fig. 1).

Mitochondrial DNA (mtDNA) introgression has also been an important area of study in these hybrid systems (*Dubey & Dufresnes, 2017*; *Hofman et al., 2012*; *Plötner et al., 2008*; *Spolsky & Uzzell, 1986*; *Spolsky & Uzzell, 1984*). For instance, despite *P.* kl. *esculentus* being expected to possess *P. ridibundus* mtDNA due to size-related behavioral differences in mating (*Berger, 1970*), many hybrids contain *P. lessonae* mtDNA, suggesting that rare backcrosses with *P. lessonae* females could have occurred (*Spolsky & Uzzell, 1986*). The transfer of mtDNA between species can influence fitness, with studies indicating that introgressed *P. ridibundus* individuals carrying *P. lessonae* mtDNA may have an adaptive advantage under hypoxic conditions potentially allowing hybrid and parental species to better cope with environmental pressures such as oxygen deficiency in polluted or warm waters (*Plötner et al., 2008*; *Tunner & Nopp, 1979*).

Human-mediated translocation of water frogs across Europe has further complicated the dynamics of *Pelophylax* kleptons. These movements, driven by activities such as the frog-leg trade and pet industry, have resulted in the introduction of species like *P. ridibundus* into new habitats, where they can outcompete and even extirpate native species

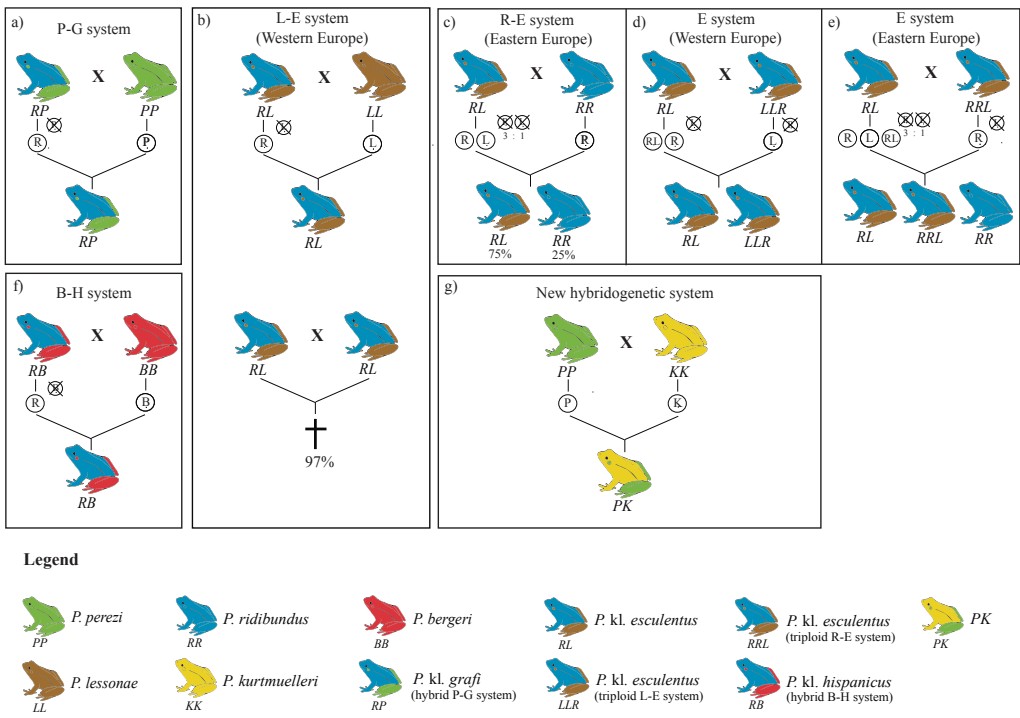

**Figure 1** Overview of the four hybridogenetic complexes of water frogs in Europe (A, B, F, G) and the intra-complex diversity according to geographical variation inducing genome exclusion (C, D, E). Each species is represented by a single color, whereas kleptons contain two colors representing their parental species. Crossed genomes are eliminated before meiosis and do not transfer into the following generation (figure modified from *Holsbeek & Jooris, 2010*).

(*Arano et al., 1995*; *Dufresnes et al., 2017*). In some cases, these introductions lead to the formation of novel kleptons between species that do not naturally coexist (*Dufresnes et al., 2017*), with significant ecological and genetic consequences for native populations (*Holsbeek & Jooris, 2010*). In regions such as Switzerland and France, native populations of *P. lessonae* and *P. perezi* have been displaced by the invasive *P. ridibundus*, which possesses a competitive advantage due to traits such as higher fecundity, larger body size, and faster growth rates (*Berger, 1970*; *Schmeller et al., 2007*; *Vorburger & Reyer, 2003*).

The genetic impact of kleptons in these systems raises significant conservation concerns. The difficulty in morphologically distinguishing kleptons from parental species complicates efforts to protect native populations (*Hauswaldt et al., 2012*). Identification of *Pelophylax* species and their hybridogenetic systems is particularly challenging due to morphological similarities, hybrid variability, and overlapping habitats. While parental species like *P. lessonae* and *P. ridibundus* can often be distinguished by size, coloration, and habitat preferences, hybrids such as *P.* kl. *esculentus* show intermediate traits, making visual identification difficult. Acoustic differences, especially in male calls, offer another means of distinguishing species, but kleptons may also exhibit overlapping vocal characteristics (*Gomes de Almeida et al., 2024*). The same applies to the *P. perezi*–*P.* kl. *grafi* (P-G) system, in which the high levels of morphological variability sometimes preclude identification

despite the apparent existence of diagnostic characters between *P. perezi* and *P.* kl. *grafi* (*Crochet et al., 1995*; *Pérez-Sorribes et al., 2018*). In any case, tadpoles of kleptons in the L-E, P-G, and other systems cannot be identified with certainty based on their morphology alone. Genetic identification, particularly through targeted molecular markers, has become essential for reliably identifying hybrids and detecting putative introductions, especially when morphological traits are ambiguous or hybrid populations are established in new regions (*Cuevas et al., 2022*).

Despite occasional reports of introduced *P. ridibundus* and *P. lessonae* specimens, most probably escaping from frog-leg farms or originating from horticultural activities (*Arano et al., 1995*), the main species of water frog in the Iberian Peninsula are *P. perezi* and *P.* kl. *grafi*. The main hybridogenetic system in this region, found surrounding the Pyrenees, is the P-G system, where the parental *P. perezi* genome is excluded during hybrid gametogenesis (*Cuevas et al., 2022*). The origins of *P.* kl. *grafi* remain unclear, but may stem from ancestral crosses between *P. perezi* and *P. ridibundus* during periods of range overlap or from crosses between *P. perezi* and *P.* kl. *esculentus* from the L-E system (*Arano & Llorente, 1995*). Identifying the precise origins of these kleptons through molecular methods is essential for unraveling their evolutionary history and supporting conservation efforts (*Dufresnes et al., 2017*). Across the French side of the Pyrenees, the composition of *Pelophylax* species is much more variable where several stable populations of introduced *P. ridibundus* and *P. r. kurtmuelleri* have been reported (*Demay et al., 2023*; *Dufresnes et al., 2017*; *Dufresnes et al., 2024a*), as well as the presence of the closely-related *P. saharicus* (*Doniol-Valcroze et al., 2021*). Additionally, the native *P. lessonae*, the kleptons *P.* kl. *esculentus*, and the newly identified klepton PK (*P. perezi* x *P. ridibundus kurtmuelleri*) occur close to the Spanish border (*Dufresnes et al., 2017*). Studies examining the natural introduction corridors some of these species may follow could offer valuable insights into the ongoing evolutionary processes between *Pelophylax* species.

In this study, we conducted a comprehensive sampling of water frogs across Catalonia, northeastern Spain, where *P. perezi* and its hybridogenetic klepton *P.* kl. *grafi* occur. We obtained an unprecedented sampling including 423 water frogs from 54 localities distributed across the main water basins. Then we used a two-step molecular identification technique, to examine the species distribution, assess the accuracy of morphological identification, and evaluate if any introduced species foreign from the P-G system occur in the region. Additionally, we explored the mitochondrial genome of *P.* kl. *grafi* hybrids to gain insights into the geographic variation and population structure in this complex. Results showed that morphological identification is highly unreliable in contrast to molecular methods. Hybrids of the P-G system are widely distributed across the entire region and, interestingly, all *P.* kl. *grafi* present *P. perezi*'s mitochondrial genomes.

## MATERIALS AND METHODS

### Sampling

A total of 423 water frogs from 54 localities were collected between January to June 2018, at the peak of the breeding season of *Pelophylax* (Fig. 2 and Table S1). In addition, samples

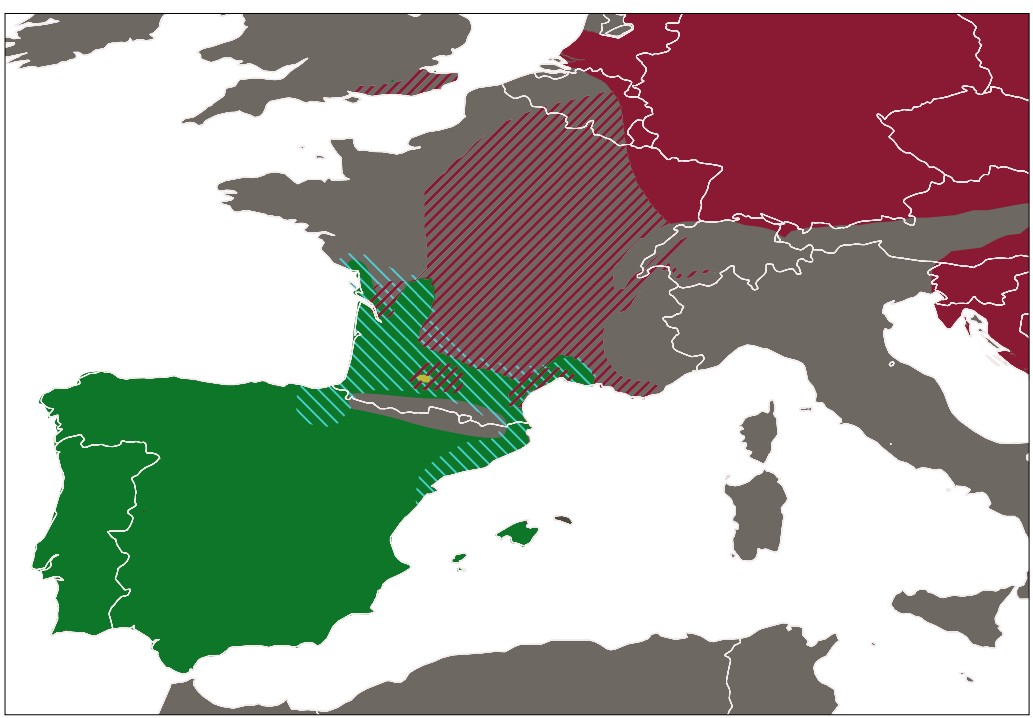

**Figure 2   Distribution range of western European water frogs.** Map shows the the distribution ranges of *Pelophylax perezi* (green), *P.* kl. *grafi* (dashed blue), *P. ridibundus* (red), the introduced range of *P. ridibundus* (dashed red), and a locality with an introduction of *P. bergeri* (yellow).

from the natural ranges of *P. ridibundus kurtmuelleri* ($n = 1$, Greece), *P. lessonae* ($n = 1$, France), and *P. ridibundus* ($n = 1$, obtained from an introduced population in Riyad, Saudi Arabia) were included. Sampling was designed to cover all main river basins within Catalonia, Spain, placing special focus in the closest basins to the French border, given the possible natural entrance of the klepton from southern France into the Iberian Peninsula. Populations connected through rivers were selected so that there was a minimum distance of 10 km between them. However, this minimum distance was not considered when localities were separated by a geographical barrier.

To ensure that the sampling effort was similar at all localities Iberian water frogs were captured over a 20-minute span. Individuals collected were identified to species level according to general morphological characteristics such as size, color, head shape, snout shape, extent of the interdigital membrane, size and shape of the metatarsal tubercle, and, especially, by the shape and distance between the proximal edges of the vomerine ridges (*Crochet et al., 1995*; *Duguet & Melki, 2003*; *Pérez-Sorribes et al., 2018*; *Plötner, Becker & Plötner, 1994*; *Plötner, 2010*). Tadpoles and doubtful specimens were classified as *Pelophylax* sp. All specimen identifications were conducted by the same people (CM-D and DF-G). For every collected specimen live stage (adult, juvenile, or tadpole) and sex (male, female, or undetermined; Table S1) were recorded, and a tissue sample was obtained by toe-clipping in case of adults or juvenile specimens, or by a five mm tail tip in case of tadpoles. Tissue

samples were preserved in absolute ethanol and stored at −20 °C. To minimize blood loss, a tissue adhesive produced from enbucrilate (Histoacryl, B. Braun Surgical) was immediately applied to the wound. All collected individuals were released at the same collecting spot after manipulation. Manipulation of all amphibian specimens followed a strict biosecurity protocol (Protocol Sanitari d'Amfibis de la Generalitat de Catalunya) established to avoid the spread and infection of emergent diseases (*Berger et al., 1998*; *Martel et al., 2013*).

Collection and manipulation of the specimens included in this work was approved by the Servei de Fauna i Flora del Departament de Territori i Sostenibilitat de la Generalitat de Catalunya (permit numbers: SF/0040, SF/0041, SF/0042, SF/0043). A special permission was acquired for collecting inside the following protected natural areas: Parc Natural del Delta de l'Ebre; Parc Natural dels Ports; Parc Natural del Montgrí, les Illes Medes i el Baix Ter; Parc Natural dels Aiguamolls de l'Empordà; Parc Natural d'Interès Nacional de l'Albera; Parc Natural de la Zona Volcànica de la Garrotxa; Espai d'Interès Natural de l'Alta Garrotxa. Regarding the sampling, study design, and sample manipulation of wildlife specimens, this study adheres to the ARROW guidelines (*Field et al., 2019*).

## Molecular species identification

Genomic DNA was extracted from the alcohol-preserved tissues following a standard high-salt protocol (*Sambrook, Fritsch & Maniatis, 1989*). A two-step molecular protocol was then performed to differentiate between pure *P. ridibundus,* pure *P. perezi,* and the klepton *P.* kl. *grafi,* as well as differentiate them from any other recognized extant species and kleptons of Western European water frogs.

The first step was based on the different cutting pattern of the PCR amplified region of the Recombination activating gene 1 (RAG1) with the restriction enzyme EcoO109I. The different number and disposition of cleavage sites between *Pelophylax* species and kleptons allow for the discrimination of *Pelophylax* into three groups: (1) One band in *P. perezi, P. lessonae* and *P. bergeri;* (2) Two bands in *P. ridibundus* and *P. kurtmuelleri;* (3) Three bands in *P.* kl. *grafi, P.* kl. *esculentus, P.* kl. *hispanicus* and PK (Fig. S1). The RAG1 nuclear region was amplified using the same primers as in *Newman et al. (2012)* MartFL1 (5′-AGC TGC AGY CAG TAC CAC AAA ATG-3′) and AMPR1rana (5′-AAT TCA GCT GCA TTT CCA ATG TC-3′) and PCR conditions were as follows: initial denaturation step at 95 °C for 5 min; 41 cycles of denaturation at 94 °C for 1 min, annealing at 59 °C for 1.5 min and elongation 72 °C for 1 min; final extension step at 72 °C for 10 min. Digestion of the amplified RAG1 bands was carried out in a final volume of 25 μL, containing 1x EcoO109I Buffer, 10 units of EcoO109I enzyme, eight μL of the amplified PCR product, and the remaining volume up to 25 μl of H2O. The digestion mix was incubated at 37° for 1.5 h and inactivated at 65° for 20 min. The final product was run on a 1.5% agarose gel for visualization and band identification.

Although, the first step already allowed for a reliable identification of all *Pelophylax* species present in Spain and large areas of Southern France (*P. ridibundus, P. perezi,* and *P.* kl. *grafi*), a second step was performed to ensure a final discrimination at the specific level for each specimen. This step relied on the different sizes of the PCR amplified fragment of the serum albumin intron-1 (SAI-1), which contains a non-LTR retrotransposon of the

chicken repeat family (RanaCR1; *Cuevas et al., 2022*; *Hauswaldt et al., 2012*). While some studies (*Cuevas et al., 2022*; *Dufresnes et al., 2024b*) have discussed potential limitations of the SAI-1 marker, particularly for distinguishing *P. lessonae* from *P. ridibundus*, this should not affect our study, as the former is not expected to occur in the study area. The SAI-1 region was amplified using the same primers as in *Hauswaldt et al. (2012)* with the following PCR conditions: Initial denaturation step of 94 °C for 1.5 min; 35 cycles of denaturation at 94 °C for 30 s, annealing at 59 °C for 40 s and elongation at 72 °C for 100 s; final extension at 72 °C for 10 min. A total of three μL of PCR product was run on a 1.5% agarose gel. As a result of the small size difference between *P. kurtmuelleri*, *P. ridibundus*, and *P. perezi*, apart from the 100 bp ladder we also loaded as a size control a mixture of amplified SAI-1 of *P. ridibundus*, *P. kurtmuelleri*, and *P. lessonae*.

## Mitochondrial DNA (mtDNA) amplification and sequencing

A region of the *cytochrome b* (*cyt-b*) mtDNA was amplified for a representative sample (194 specimens) of all the specimens identified as kleptons including at least one sample from each locality where they were identified, to investigate the origin of their maternal DNA and the levels of mtDNA genetic diversity (Table S1). The *cyt-b* region of 647 bp was amplified using primers CYTB_PeloF (5′-ATG ATG AAA CTT TGG CTC C-3′) and CYTB_PeloR (5′-TCT GGC TTA ATG TGG GGG-3′) with the following PCR conditions: Initial denaturation step of 94 °C for 5 min; 35 cycles of denaturation at 94 °C for 30 s, annealing step at 56 °C for 45 s and elongation at 72 °C for 80 s; final extension step at 72 °C for 5 min. Sequencing of the PCR products was carried out by Macrogen Europe.

## Phylogenetic and network analyses

For all 194 sequences, chromatographs were checked manually, assembled, and edited using Geneious v.9.0.5 (Biomatters Ltd.). Sequences were translated into amino acids and no stop codons were observed. A haplotype network was inferred for the 194 *cyt-b* sequences obtained in the present work using the TCS statistical parsimony network approach (*Clement, Posada & Crandall, 2000*) implemented in the program PopART (*Leigh & Bryant, 2015*). A subset of 22 specimens was then selected representing each retrieved haplotype for phylogenetic analyses (see Results). The 22 representative sequences were aligned together with a set of sequences downloaded from GenBank, including most of the described species of *Pelophylax* and a representative of *Lithobates catesbeianus* that was used to root the phylogenetic tree (Table S2; sequences from *Dubey & Dufresnes, 2017*; *Lymberakis et al., 2007*). The assembled dataset was aligned using MAFFT v.6 (*Katoh & Standley, 2013*) with default parameters. Phylogenetic analyses were performed using maximum likelihood (ML) and Bayesian inference (BI) methods. A GTR+G substitution model was selected under the Akaike information criterion (*Akaike, 1973*) using jModelTest v.2.10 (*Posada, 2008*), and ML phylogenetic analyses were conducted with RAxML v.7.4.2 (*Stamatakis, 2006*) with 100 random starting trees and 1,000 bootstrap replicates. BI analyses were performed with MrBayes v.3.2.6 (*Ronquist et al., 2012*). Two simultaneous runs were performed with four chains per run for $2 \times 10^6$ generations sampling every 200 generations. The standard deviation of the split frequencies between the two runs and the
potential scale reduction factor (PSRF) diagnostic were examined and convergence and stationarity were ensured (ESS > 200) using Tracer v.1.6 (*Rambaut & Drummond, 2013*). We conservatively discarded the first 25% of trees as burn-in.

## RESULTS

### Field sampling and preliminary morphological identification

Our sampling effectively covered the entire known distribution range of the genus *Pelophylax* in Catalonia (Fig. 2), including all major river basins. The final dataset comprised 423 water frogs spanning through 54 localities (Table 1). From all 423 *Pelophylax*, 125 were males, 65 females, 164 juveniles, and 69 tadpoles (Table 2). According to morphological identification, 192 specimens were classified as *P. perezi*, 79 as *P. kl. grafi*, and 152 could not be identified (mainly juveniles and tadpoles; Table 2 and S1). Genetic analyses revealed frequent misidentification of *P. kl. grafi* based on morphology in both sexes (28 adult *P. kl. grafi* males and 23 females were morphologically assigned to *P. perezi*). Similarly, 42 juveniles of *P. kl. grafi* were misidentified as *P. perezi*. Conversely, morphological assignment of *P. kl. grafi* was more reliable, with only four *P. perezi* adults (three males, one female) misidentified as *P. kl. grafi* (Table 2).

### Molecular identification and geographic distribution

The two-step molecular identification method allowed for the unambiguous identification of all 423 sampled water frogs, including all tadpoles (Table 2 and Fig. 3, Figs. S2, and S3). Interestingly, despite the high number of specimens analyzed from a wide range of localities across Catalonia, only two species of water frogs were found, *P. perezi* and *P. kl. grafi*. As shown in Table 1, 23 out of the 54 populations sampled presented both *P. perezi* and *P. kl. grafi* in sympatry, with 17 populations including only representatives of *P. perezi* and 14 populations only representatives of *P. kl. grafi*. While both species are widely distributed across Catalonia, a relatively large region between Barcelona and the Ebro Delta seems devoid of *P. kl. grafi* (*i.e.,* populations P4, P12, P14-P17, P21-P23 add up to 53 *P. perezi* and no *P. kl. grafi*; Fig. 3). Localities with exclusively *P. kl. grafi* tended to have small sample sizes (1–5 specimens), with only three localities (P25, P39, and P46) having larger sample sizes of 8, 16, and 16 specimens, respectively. No clear geographic pattern emerges in the distribution of the 14 populations containing only *P. kl. grafi* (Fig. 3), and the presence of *P. perezi* at low frequencies cannot be ruled out due to the limited sample sizes.

### Mitochondrial DNA (mtDNA) analyses

A total of 194 specimens of *P. kl. grafi* were selected for molecular analyses, including at least one representative from each one of the 37 localities with *P. kl. grafi* specimens representing 78.5% of the total number of *P. kl. grafi* specimens identified in this work (Table 1). The selected specimens were sequenced for the *cyt-b* mitochondrial gene in order to know: (1) The mitochondrial composition of the kleptons; (2) the level of mtDNA genetic variability, and (3) the geographic distribution of the different haplotypes.

The results of the phylogenetic analyses on the *cyt-b* indicate that all 194 *P. kl. grafi* analyzed had *P. perezi* mtDNA (Fig. 4). A total of 22 unique haplotypes (H) were identified,

**Table 1 Summary of the total specimens collected in each of the 54 populations.** For each population, we specify the number of *P. perezi* and *P.* kl. *grafi* specimens, the number of *P.* kl. *grafi* sequenced in this study, the haplotypes present in each population, and the number of individuals containing each haplotype.

| Population code | Total specimens | *P. perezi* | *P.* kl. *grafi* | | Haplotype (n) |
|---|---|---|---|---|---|
| | | | Total | Sequenced | |
| P1 | 10 | 2 | 8 | 8 | H18 (4); H22 (4) |
| P2 | 1 | 1 | – | – | – |
| P3 | 1 | 1 | – | – | – |
| P4 | 3 | 3 | – | – | – |
| P5 | 10 | 1 | 9 | 8 | H1 (5); H2 (1); H20 (1); H22 (1) |
| P6 | 2 | – | 2 | 2 | H1 (1); H16 (1) |
| P7 | 1 | – | 1 | 1 | H1 (1) |
| P8 | 1 | – | 1 | 1 | H1 (1) |
| P9 | 1 | – | 1 | 1 | H1 (1) |
| P10 | 1 | – | 1 | 1 | H1 (1) |
| P11 | 5 | 4 | 1 | 1 | H12 (1) |
| P12 | 9 | 9 | – | – | – |
| P13 | 2 | 1 | 1 | 1 | H22 (1) |
| P14 | 3 | 3 | – | – | – |
| P15 | 6 | 6 | – | – | – |
| P16 | 8 | 8 | – | – | – |
| P17 | 5 | 5 | – | – | – |
| P18 | 5 | 4 | 1 | 1 | H19 (1) |
| P19 | 29 | 3 | 26 | 12 | H1 (11); H18 (1) |
| P20 | 22 | 22 | – | – | – |
| P21 | 8 | 8 | – | – | – |
| P22 | 9 | 9 | – | – | – |
| P23 | 2 | 2 | – | – | – |
| P24 | 9 | 2 | 7 | 7 | H3 (7) |
| P25 | 8 | – | 8 | 8 | H1 (1); H22 (7) |
| P26 | 23 | 1 | 22 | 11 | H1 (11) |
| P27 | 13 | 6 | 7 | 7 | H1 (5); H22 (2) |
| P28 | 2 | 1 | 1 | 1 | H22 (1) |
| P29 | 2 | 1 | 1 | 1 | H22 (1) |
| P30 | 2 | 1 | 1 | 1 | H1 (1) |
| P31 | 12 | 2 | 10 | 8 | H1 (3); H4 (1); H8 (1); H11 (1); H22 (2) |
| P32 | 15 | 3 | 12 | 9 | H11 (2); H13 (4); H22 (3) |
| P33 | 14 | 1 | 13 | 9 | H1 (5); H15 (2); H22 (2) |
| P34 | 6 | 6 | – | – | – |
| P35 | 6 | 2 | 4 | 4 | H1 (4) |

*(continued on next page)*
**Table 1** (*continued*)

| Population code | Total specimens | *P. perezi* | *P.* kl. *grafi* Total | *P.* kl. *grafi* Sequenced | Haplotype (n) |
|---|---|---|---|---|---|
| P36 | 25 | 15 | 10 | 10 | H1 (5); H5 (1); H8 (1); H9 (1); H17 (1); H21 (1) |
| P37 | 6 | 1 | 5 | 5 | H1 (1); H7 (2); H8 (2) |
| P38 | 5 | 1 | 4 | 4 | H1 (3); H22 (1) |
| P39 | 16 | – | 16 | 16 | H1 (16) |
| P40 | 1 | – | 1 | 1 | H22 (1) |
| P41 | 1 | – | 1 | 1 | H22 (1) |
| P42 | 2 | – | 2 | 2 | H1 (1); H22 (1) |
| P43 | 2 | – | 2 | 2 | H10 (2) |
| P44 | 5 | 2 | 3 | 3 | H1 (2); H6 (1) |
| P45 | 8 | 8 | – | – | – |
| P46 | 16 | – | 16 | 8 | H22 (8) |
| P47 | 11 | 11 | – | – | – |
| P48 | 11 | 6 | 5 | 4 | H22 (4) |
| P49 | 3 | – | 3 | 3 | H22 (3) |
| P50 | 3 | 2 | 1 | 1 | H22 (1) |
| P51 | 7 | 7 | – | – | – |
| P52 | 2 | 2 | – | – | – |
| P53 | 38 | 3 | 35 | 26 | H1 (9); H16 (17) |
| P54 | 5 | – | 5 | 5 | H1 (3); H14 (2) |

**Table 2** **Morphological and genetic identification of the 423 individuals collected in this study.** Specimens are divided in four groups according to life stage: adults (males and females), juveniles, and tadpoles. In brackets, detected morphological identification errors and undetermined specimens using the two-step molecular protocol. Superscript "g" and "p" refer to *P.* kl. *grafi* and *P. perezi*, respectively.

| Life stage | N | Morphology *P. perezi* | Morphology *P.* kl. *grafi* | Morphology Undetermined | Genetics *P. perezi* | Genetics *P.* kl. *grafi* | Genetics Undetermined |
|---|---|---|---|---|---|---|---|
| **Males** | 125 | 70 (28[g]) | 50 (3[p]) | 5 (5[p]) | 50 | 75 | 0 |
| **Females** | 65 | 42 (23[g]) | 21 (1[p]) | 2 (2[p]) | 22 | 43 | 0 |
| **Juveniles** | 164 | 80 (42[g]) | 8 | 76 (24[p]; 52[g]) | 62 | 102 | 0 |
| **Tadpoles** | 69 | 0 | 0 | 69 (42[p]; 27[g]) | 42 | 27 | 0 |

with H1, H16, and H22 being the most frequent, represented by 91, 18, and 44 specimens, respectively. As shown in Table 1 and Fig. 5, the most common haplotype (H1) is distributed throughout the entire study area, except for the northwesternmost part, where only H22 was found. Aside from the extreme northwest, H22 is also widely distributed around Barcelona stretching to the north but seems absent from the southern half of Catalonia (15 specimens analyzed from three different populations, P18-P19, P43). The third most common haplotype was restricted to two geographically close populations (P6, P53) from the Montnegre i el Corredor Natural Park. While some populations with a relatively high number of sequenced specimens presented a single haplotype (P39, P11, P24, P46),

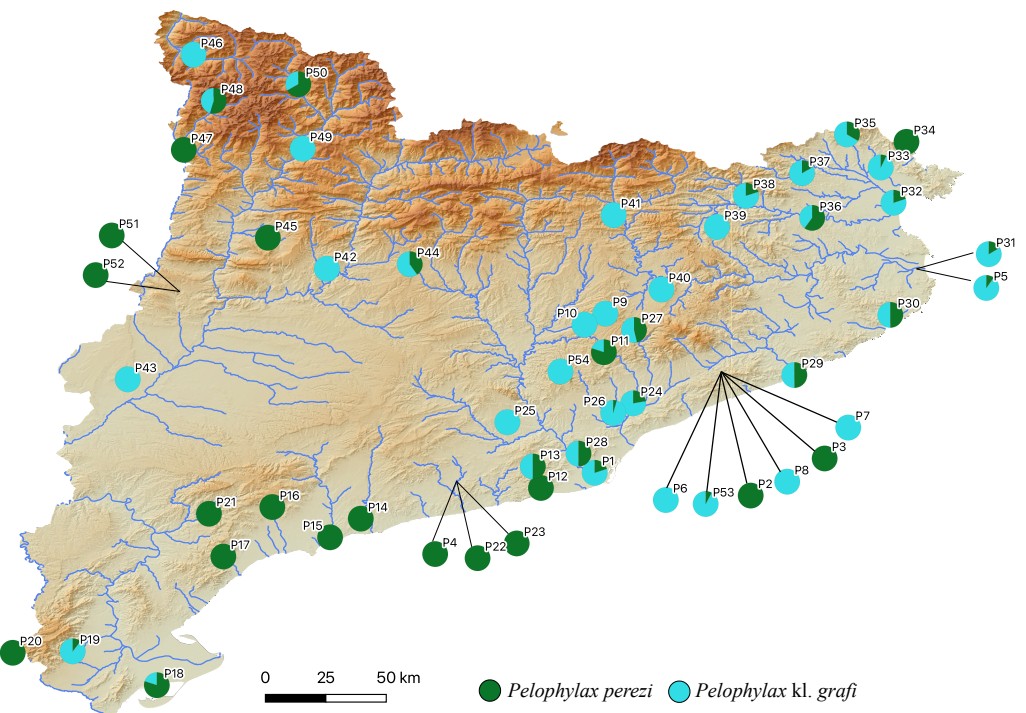

**Figure 3** **Distribution map of *P. perezi* (green) and *P.* kl. *grafi* (blue) in Catalonia along the main river basins.** Pie diagrams represent the species' proportions in each population.

other populations presented higher haplotype diversity with the highest being P36 (six haplotypes), P31 (five haplotypes), P5 (four haplotypes), and P32-P33 and P37 (three haplotypes). Interestingly, and in contrast to the situation in northwestern Catalonia, all populations with three or more haplotypes were restricted to northeastern Catalonia (Fig. 5).

## DISCUSSION

### Identification and distribution of *Pelophylax* in Catalonia

In the present study, we have generated a comprehensive and exhaustive sampling of *Pelophylax* water frogs across Catalonia. This region serves as a natural connection between the Iberian Peninsula and the rest of Europe due to the lower mountain ranges found in the easternmost side of Pyrenees. Not surprisingly, this area has been a key corridor for the dispersal of several amphibian genera into Europe, such as *Salamandra* (*Gippner et al., 2024*), *Alytes* (*Ambu et al., 2023*; *Dufresnes & Hernandez, 2021*; *Dufresnes & Martínez-Solano, 2020*), or *Pelodytes* (*Dufresnes et al., 2020*), but it may have also facilitated the entry of other groups, including *Triturus marmoratus* (*Kazilas et al., 2024*) or *Discoglossus pictus* (*Dufresnes et al., 2020*), into Iberia. Given its potential role as a natural dispersal route for *Pelophylax* species and kleptons from France into the Iberian Peninsula, this region provides an ideal setting to investigate the dynamics of water frogs in Iberia (*Arano et*

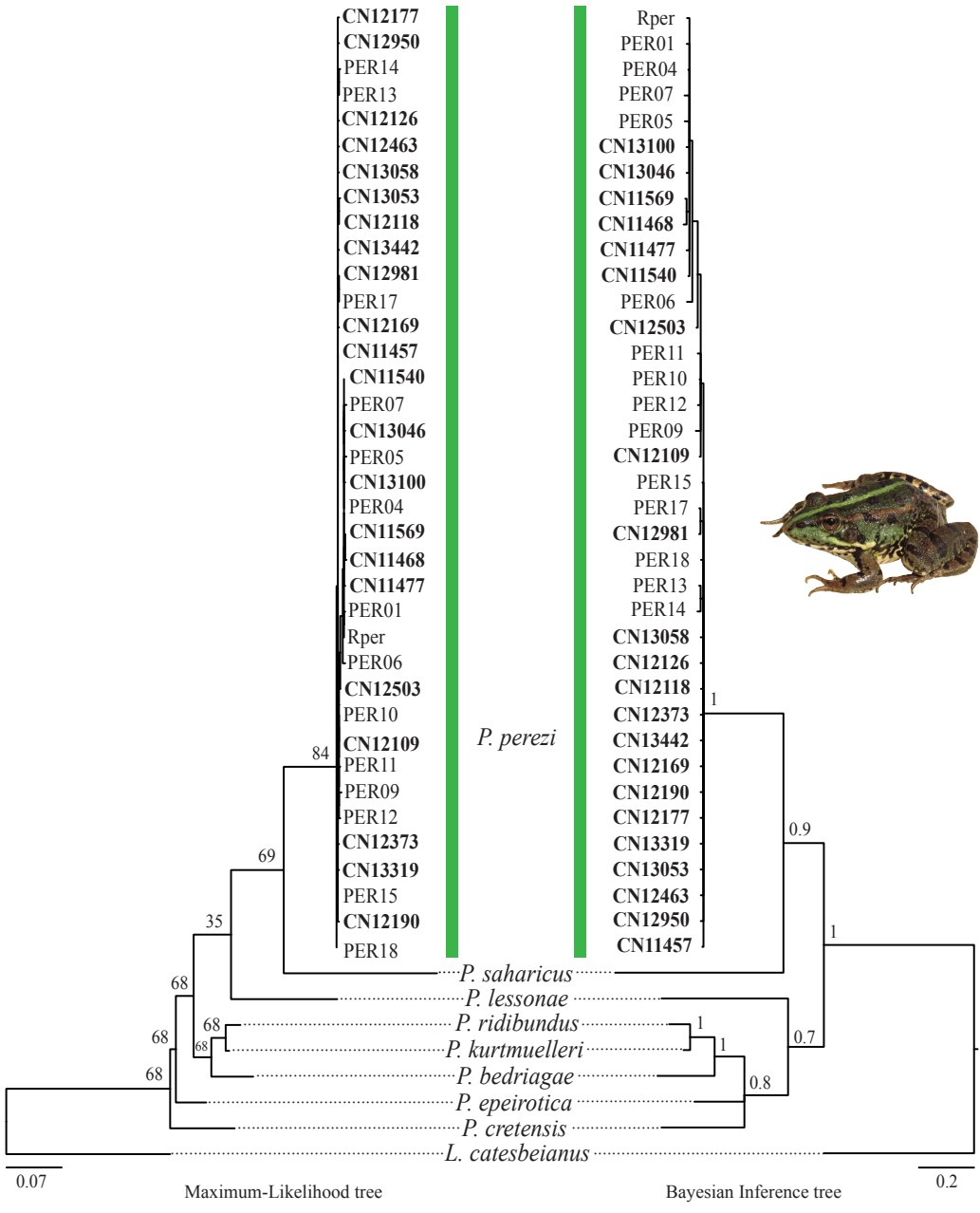

**Figure 4** ***Pelophylax* kl. *grafi* phylogenetic trees.** *Cytochrome b* maximum-likelihood (left) and Bayesian inference (right) phylogenetic trees of some selected *Pelophylax* species including all unique haplotypes found in *P*. kl. *grafi* specimens, allowing the identification of the mitochondrial genome origin of 194 *P*. kl. *grafi*. Sequences in bold are from this study.

*al., 1995*; *Arano & Llorente, 1995*). Here, we have provided reliable identification for 423 specimens through the implementation of a two-step, cost-effective molecular protocol and showed that, despite previous reports on the presence of *P. idibundus* and the hybrid *P*. kl. *esculentus* in the Iberian Peninsula (*Arano et al., 1995*; *Holsbeek & Jooris, 2010*), the

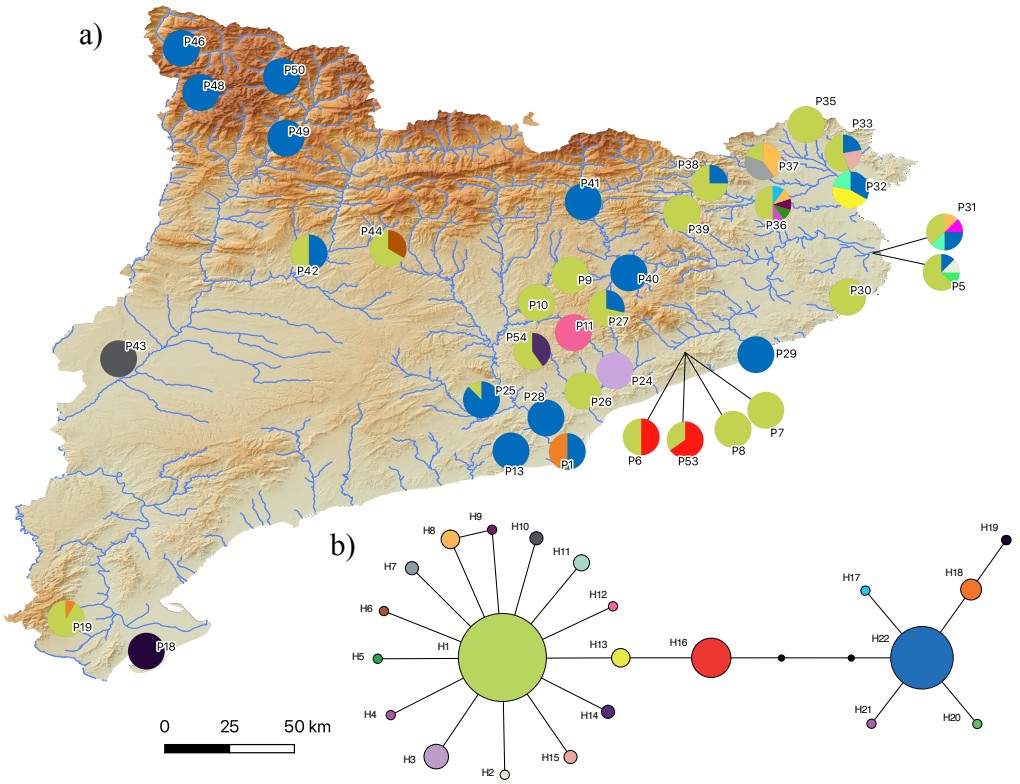

**Figure 5** *Pelophylax* **kl.** *grafi* **mitochondrial haplotypes.** (A) Map of the 22 haplotypes found in *P.* kl. *grafi* distributed across Catalonia. Pie diagrams represent haplotype proportions among *P.* kl. *grafi* individuals at each locality. (B) Mitochondrial network showing haplotype diversity corresponding to *P. perezi* mitochondrial genomes found in 194 hybrid specimens sequenced for the *cytochrome b*. Colors represent mitochondrial lineages.

only species found in this work were the autochthonous species *P. perezi* and the hybrid *P.* kl. *grafi* of the P-G hybridogenetic system (Fig. 1A).

The usage of a molecular protocol has been vital for the proper identification of *P. perezi* and *P.* kl. *grafi*, as the morphological identification of both species has proven unreliable in adults (incorrect identification was as high as 26% in males and 38% in females), and nearly impossible in juveniles (26% of misidentifications and 46% of undetermined specimens; Table 2). Most identification errors stem from *P.* kl. *grafi* being incorrectly identified as *P. perezi*, making the primary diagnostic character used to differentiate between *P. perezi* and *P.* kl. *grafi* (the distance between the vomerine ridges) useful only in clear cases—either when the vomerine ridges are distinctly separated, as in *P. perezi,* or when they are clearly in contact, as in *P.* kl. *grafi*. This situation contrasts with the L-E system of *P.* kl. *esculentus* in which, morphological identification of the klepton is more reliable. For instance, *Plötner (2010)* was able to assign in the field 90% of adult individuals correctly to either parental (*P. lessonae* or *P. ridibundus*) and the hybrid (*P.* kl. *esculentus*) based on morphological characters, particularly the size and shape of the metatarsal tubercle. Additionally, the

usage of this molecular protocol has allowed for the identification of tadpoles ($n = 69$), for which there are no morphological diagnostic characters available.

*Arano & Llorente (1995)* reported the presence of *P.* kl. *grafi* in six localities widely distributed across Catalonia. In the present study, we observed an increase in the abundance of *P.* kl. *grafi* in those previously reported localities (Table 1) and expanded its known distribution to nearly the entire study area, with occurrences in 37 localities (Table 1). The only exception was the southeastern Catalonian coast, where no *P.* kl. *grafi* were recorded despite intensive sampling efforts (Fig. 3 and Table 1). The absence of kleptons in these populations was previously proposed to be explained by a relatively recent entrance of the complex from Southern France into the Iberian Peninsula (*Arano & Llorente, 1995*). However, the relatively wide distribution of *P.* kl. *grafi* already reported 23 years ago (*Arano & Llorente, 1995*), the high dispersal capability of these frogs (as evidenced by the widely distributed mitochondrial haplotypes; Fig. 5), and the high abundance of *P.* kl. *grafi* in population P19 further south (accounting for 90% of all sampled specimens; Fig. 3 and Table 1) suggest that other factors may play a role in the absence of *P.* kl. *grafi* from this large area of Catalonia. One such factor could be the lack of waterborne corridors (such as rivers or streams) reaching that region, which may impose a geographic barrier impeding the dispersal of the klepton.

## Origin and maintenance of *P.* kl. *grafi*

The origin and age of arrival of *Pelophylax* kl. *grafi* into the Iberian Peninsula is still unknown. *Arano & Llorente (1995)* considered two entry paths for the P-G system in the Iberian Peninsula, one across the central Pyrenees following the Segre-Ebro River system and another one across the eastern Pyrenees, following the Llobregat River. In the present study, we have also found populations with *P.* kl. *grafi* (P46, P48-P50; Fig. 3) in northwestern Catalonia, suggesting that colonization through the river basins directly connecting with France through the Val d'Aran (Aran Valley) could have also been possible. Further studies evaluating the nuclear *P. ridibundus* genome in *P.* kl. *grafi* are needed to better understand how and when this genome entered the Iberian Peninsula.

The lack of *P. ridibundus* in Catalonia may give some clues on how many times this particular hybridogenetic system has appeared. In the L-E system, mattings between two *P.* kl. *esculentus* almost invariably leads to *P. ridibundus* tadpoles with developmental difficulties, resulting in death before reaching sexual maturity (*Berger, 1967*; *Graf & Polls-Pelaz, 1989*; *Vorburger, 2001a*; *Vorburger, 2001b*; *Vorburger, 2001c*) (Fig. 1B). This is the consequence of the accumulation of recessive deleterious mutations due to the Muller's ratchet (*Muller, 1964*) in the hemiclonaly inherited *P. ridibundus* genome (*Holsbeek & Jooris, 2010*; *Lehtonen et al., 2013*). However, as demonstrated by *Vorburger (2001b)*, experimental crossings between *P.* kl. *esculentus* pairs with different hemiclonal *P. ridibundus* genomes produce viable *P. ridibundus* offspring with no signs of reduced fitness. Moreover, it has also been reported rare cases (3%) of all-female offspring viability between crossings of *P.* kl. *esculentus* bearing the same clonal genome (*Vorburger, 2001b*). In such cases, recombination of the clonal genomes they inherited from the two parental hybrids seems to occur and, after mating with syntopic *P. lessonae*, produce new hemiclones

with novel combinations of alleles, thus providing an escape from the evolutionary dead-end they were trapped in. The lack of *P. ridibundus* in the present study suggests that hybrids in the P-G system are not able to produce viable offspring, pointing towards a single origin for the hemiclonal *P. ridibundus* genome transmitted by *P.* kl. *grafi*. Multiple origins of the P-G hybridogenetic system would involve more diversity and a higher probability for successful offspring resulting from a crossing between two hybrids.

In populations where hybrids and *P. perezi* coexist, *P.* kl. *grafi* tends to be more abundant than *P. perezi* (Fig. 3). This result aligns with the reproductive strategy of hybridogenetic complexes, where offspring between hybrids and pure specimens will always result in more hybrids. Additionally, hybrid females are larger than *P. perezi* females, providing a further reproductive advantage over *P. perezi*. However, this increased reproductive success of *P.* kl. *grafi* poses an evolutionary conundrum that should, theoretically, lead to the collapse of water frogs of the P-G system (*Vorburger & Reyer, 2003*). If the reproductive advantage of the klepton over the parental species were to drive *P. perezi* to extinction, it would also result in the demise of the klepton due to the non-viability of hybrid-hybrid crossings. Nevertheless, the persistence of hybrid populations over at least 30 years (*Arano et al., 1995*) suggests that some form of equilibrium might exist within the system. This ecological stability could be achieved through sexual preferences in mating, similar to those observed in the L-E system (*Bove, Milazzo & Barbuti, 2014*; *Skierska et al., 2023*). Empirical and theoretical studies have shown that the stability of L-E populations can be maintained by specific female mate choice, with *P. lessonae* females preferentially mating with *P. lessonae* males over *P.* kl. *esculentus* (*Bove, Milazzo & Barbuti, 2014*; *Skierska et al., 2023*). Moreover, female mate choice can facilitate the diffusion of deleterious mutations within the system (*Bove, Milazzo & Barbuti, 2014*). However, further experimental studies and long-term monitoring of water frog populations are necessary to determine whether this ecological stability exists or if *P. perezi* is gradually declining. Future monitoring should especially focus on localities where only *P.* kl. *grafi* has been reported to assess the viability of these populations and to better understand how full *P.* kl. *grafi* populations can persist.

Since the origin of the klepton remains unknown, it is not possible to consider the *P.* kl. *grafi* as either a native taxon or the result of anthropogenic intervention. However, the stability of the P-G system could be significantly threatened by the recent introduction of other *Pelophylax* species to the system. It has been shown that the introduction of pure *P. ridibundus* specimens can break the stability of the L-E system leading to the collapse of the entire population (*Bove, Milazzo & Barbuti, 2014*). Several stable populations of introduced *P. ridibundus*, *P. kurtmuelleri*, *P. lessonae*, and *P.* kl. *esculentus* as well as the recently identified klepton PK (*i.e.,* P. perezi x *P. kurtmuelleri*) have already been reported in southern France, very close to the Spanish border (Fig. 1; *Dufresnes et al., 2017*; *Dufresnes et al., 2024a*; *Demay et al., 2023*). While the Pyrenees seem to be acting as a geographic barrier to the dispersal of these species into the Iberian Peninsula (*i.e.,* no evidence of any of those species has been reported in this study), it is only a matter of time until some of these species reach the Iberian Peninsula, probably leading to the destabilization of the P-G system.

### Origin and genetic variability of the mtDNA of *P*. kl. *grafi* in Catalonia

According to the reconstructed phylogenies (Fig. 4), all 194 sequenced *P*. kl. *grafi* specimens present *P. perezi* mtDNA. This could indicate that the first hemiclone occurred between a female *P. perezi* and a male *P. ridibundus* and therefore the *P*. kl. *grafi* never had mtDNA of *P. ridibundus* or *P. lessonae* (*i.e.,* a possibility if the origin was from the *P*. kl. *esculentus*, which has been shown to have mainly *P. lessonae* mtDNA; *Plötner et al., 2008*). However, studies conducted on the L-E system of *P*. kl. *esculentus* suggest that, due to size-related behavioral factors, natural primary hybridizations between *P. ridibundus* and *P. lessonae* are more likely to occur between large *P. ridibundus* females and small *P. lessonae* males, rather than the reciprocal pairing (*Berger, 1970*; *Borkin & Tikhenki, 1979*). Another possibility is that mattings occur between *P. perezi* females and *P*. kl. *grafi* males. Such matings produce *P*. kl. *grafi* lineages carrying *P. perezi* mtDNA. In the absence of the parental species *P. ridibundus* or *P*. kl. *esculentus* (which may have *P. ridibundus* or *P. lessonae* mtDNA), the transfer of *P. perezi* mtDNA into the hybridogenetic lineage becomes irreversible. As previously suggested in the L-E system (*Plötner et al., 2008*), the mtDNA of the local *P. perezi* populations may offer a selective advantage over *P. ridibundus* mtDNA in withstanding the often hypoxic and polluted environments where *P. perezi* and *P*. kl. *grafi* populations are commonly found. Similar to *P. ridibundus* tadpoles introgressed with *P. lessonae* mtDNA, it is plausible that *P*. kl. *grafi* individuals carrying *P. perezi* mtDNA are better adapted to survive under hypoxic conditions.

Judging by the high levels of mtDNA genetic variability found in *P*. kl. *grafi* (22 different haplotypes; Fig. 5), crossings between male *P*. kl. *grafi* and female *P. perezi* must be a frequent phenomenon, facilitating the continuous incorporation of new *P. perezi* mitochondrial genomes into the hybrids. The large distribution range of certain haplotypes, such as H1 and H22, suggests a high dispersal capability, with gene flow being possible even between very distant localities (Fig. 5). This could be favored by the fact that *Pelophylax* is a genus of highly aquatic frogs capable of establishing populations in a variety of habitats and tolerating a wide range of environmental conditions, including salinity, contamination, human activities, and eutrophication. The only significant limiting factor for their presence is the permanent absence of water (*Arano & Llorente, 1995*; *Barbadillo et al., 1999*). It is noteworthy that some populations have up to six different haplotypes, most likely acquired through *in situ* crossings within sympatric *P. perezi*. However, since we did not sequence any *P. perezi* mtDNA we cannot ensure that the geographic distribution of the genetic variability found in *P*. kl. *grafi* (Fig. 5) mirrors the phylogeography of *P. perezi* in Catalonia.

## CONCLUSIONS

In this study, we have conducted a comprehensive sampling of water frogs of the P-G hybridogenetic system in Catalonia, one of the possible dispersal routes of this klepton into the Iberian Peninsula. Despite previous records of the presence of the invasive *P. ridibundus* and *P*. kl. *esculentus* in the Iberian Peninsula, these species were not detected in the present study. Additionally, we have proven that, although previous publications on the morphological differentiation between *P. perezi* and *P*. kl. *grafi*

(*Crochet et al., 1995*; *Pérez-Sorribes et al., 2018*), morphological identification of these species is highly unreliable.

Given the large sampling size (423 specimens collected across 54 localities) and the high proportion of *P.* kl. *grafi* observed, the absence of pure *P. ridibundus* in these systems suggests that crosses between *P.* kl. *grafi* individuals do not produce viable offspring (*Berger, 1967*; *Graf & Polls-Pelaz, 1989*; *Vorburger, 2001a*), and that polyploid all-hybrid populations, similar to the Eastern European E-system of *P.* kl. *esculentus* (Fig. 1E), do not exist in Catalonia. These results suggest a single colonization event of the Eastern Iberian Peninsula by this klepton. Since *P. perezi* is essential for the reproduction of *P.* kl. *grafi* and their offspring are always hybrid, a decline in *P. perezi* populations is expected. However, since *P. perezi* cannot completely disappear, as this would trigger the extinction of *P.* kl. *grafi* as well (*Lehtonen et al., 2013*), some sort of ecological stability should be occurring, possibly driven by female mate choice. Interestingly, all analyzed *P.* kl. *grafi* individuals carried *P. perezi* mitochondrial genomes, and subsequent mitochondrial analyses supported that crossings between *P. perezi* females and *P.* kl. *grafi* males are frequent, resulting in 22 mitochondrial haplotypes in *P.* kl. *grafi*. Finally, further research such as conducting experimental crossings to determine the viability of *P. perezi* and *P.* kl *grafi* offspring, evaluating mate choice and the viability of hybrid crossings, or studying the clonal *P. ridibundus* genome, among others, will be vital to better understand the P-G hybridogenetic system, as most conclusions of this study have been derived through comparison to the better-studied L-E system.

## ACKNOWLEDGEMENTS

We are grateful to Petros Lymberakis for providing samples from Greece; we are very grateful to Josep Francesc Pannon, Mireia Vila and Mariona Picart (Parc del Montnegre i el Corredor), Pedro Torres and Vanessa Gomez (Parc del Foix), Rafael Gonzalez, Josep Calaf and Daniel Pons (Parc del Garraf), Lluïsa Bachs (Parc Natural de Sant Llorenç del Munt i l'Obac), Albert Martínez and Quim Soler (CRARC), Joan Mayné and Laura Olid (Centre de Recuperació de Fauna Salvatge de Torreferrussa), Joan Budó (Centre de Recuperació de Tortugues de l'Albera), Francesc Fanalias (Consorci de l'Alta Garrotxa), Antoni Curcó (Parc Natural del Delta de l'Ebre), Ona Alay and Bartomeu Borràs (Parc Natural de l'Albera), Santi Ramos (Parc Natural del Montgrí, les Illes Medes i el Baix Ter), Joan Mestre (Parc Natural dels Ports), Emili Basols and Francesc Xavier Puig (Parc Natural de la Zona Volcànica de la Garrotxa), and Rosa Marsol and Roger (Centre de Recuperació de Fauna de Pont de Suert) for their collaboration and for facilitating sampling in the respective Natural Parks or wildlife centers. Thanks also to Associació mediambiental La Sínia, Coloma Tomàs, Carme Schouten, Albert Montori, Luis Guilherme Sousa, Josep Marí, and Pep Xarles for their collaboration in several aspects of the project.

### Funding
This project was funded by Fundació Barcelona Zoo (Pric-2019 and Amphibian Project 2024), MCIN/AEI/10.13039/501100011033/ (CGL2015-70390-P) and by FEDER "A way to make Europe". The funders had no role in study design, data collection and analysis, decision to publish, or preparation of the manuscript.

### Grant Disclosures
The following grant information was disclosed by the authors:
Fundació Barcelona Zoo (Pric-2019 and Amphibian Project 2024), MCIN/AEI/10.13039/501100011033/ (CGL2015-70390-P).
FEDER "A way to make Europe".

### Competing Interests
The authors declare there are no competing interests.

### Author Contributions
- Bernat Burriel-Carranza conceived and designed the experiments, performed the experiments, analyzed the data, prepared figures and/or tables, authored or reviewed drafts of the article, and approved the final draft.
- Carolina Molina-Duran conceived and designed the experiments, performed the experiments, analyzed the data, prepared figures and/or tables, authored or reviewed drafts of the article, and approved the final draft.
- Karin Tamar conceived and designed the experiments, performed the experiments, analyzed the data, authored or reviewed drafts of the article, and approved the final draft.
- Laia Pérez-Sorribes performed the experiments, prepared figures and/or tables, authored or reviewed drafts of the article, and approved the final draft.
- Jhulyana López-Caro performed the experiments, authored or reviewed drafts of the article, and approved the final draft.
- Mar Cirac performed the experiments, authored or reviewed drafts of the article, and approved the final draft.
- Daniel Fernandez-Guiberteau conceived and designed the experiments, performed the experiments, authored or reviewed drafts of the article, and approved the final draft.
- Salvador Carranza conceived and designed the experiments, performed the experiments, authored or reviewed drafts of the article, and approved the final draft.

### Field Study Permissions
The following information was supplied relating to field study approvals (i.e., approving body and any reference numbers):
Collection and manipulation of the specimens included in this work was approved by the Servei de Fauna i Flora del Departament de Territori i Sostenibilitat de la Generalitat de Catalunya (permit numbers: SF/0040, SF/0041, SF/0042, SF/0043).

## DNA Deposition

The following information was supplied regarding the deposition of DNA sequences:

CytB sequences are available at GenBank and the accession numbers are available in the Supplemental Files. Raw sequence data is also available in the Supplemental Files.

## Data Availability

Raw data and a list of all specimens with their identification are available in the Supplemental Files.

## Supplemental Information

Supplemental information for this article can be found online at http://dx.doi.org/10.7717/peerj.19895#supplemental-information.

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
