# Peer review of "Distribution and evolution of the western European water frogs (genus Pelophylax) from Catalonia, northeastern Spain"

_PeerJ, doi:10.7717/peerj.19895_

## Round 0.1 · original submission · Major Revisions

Thank you for your submission to PeerJ.

Please change your manuscript as reviewers' comments.

Please check the annotated pdf from one of reviewers.

Reviewer 1 ·

Basic reporting

First round of review of the article entitled “Distribution and evolution of the western European water frogs (genus Pelophylax) from Catalonia, northeastern Spain” submitted by Burriel-Carranza et al. to PeerJ.
This study is well motivated and is in line with current research and questions about the Water frog systems, their distribution as well as invasion and conservation issues in this genus. It follows previous studies on other geographic areas and fills substantial knowledge gaps about distribution and colonization history of the genus Pelophylax in Western Europe and notably in Iberia. The present work also opens perspectives to go deeper in the understanding of the buildup and functioning of Pelophylax hybridogenetic systems. I feel that this work is a substantial piece of the vast Pelophylax puzzle and will be happy to see it published after the addition of substantial precisions and rewriting. After responding to PeerJ’ Editorial Criteria, I’ve made substantial comments about additions and clarifications to the introduction, the methods and the results as well as ways to improve the clarity of the discussion and the conclusion of the study below.
1. Basic Reporting
Clear and unambiguous, professional English used throughout. : OK
Literature references, sufficient field background/context provided : OK see below for a few comments
Professional article structure, figures, tables. Raw data shared : OK see below for a few comments

Figures should be relevant to the content of the article, of sufficient resolution, and appropriately described and labeled : OK see below for a few comments

All appropriate raw data have been made available in accordance with our Data Sharing policy : OK see below for a few comments
Self-contained with relevant results to hypotheses : OK

Experimental design

2. Experimental design
Original primary research within Aims and Scope of the journal : OK
Research question well defined, relevant & meaningful. It is stated how research fills an identified knowledge gap : OK
Rigorous investigation performed to a high technical & ethical standard : OK
Methods described with sufficient detail & information to replicate : OK

Validity of the findings

3. Validity of the Findings
Impact and novelty not assessed. Meaningful replication encouraged where rationale & benefit to literature is clearly stated : OK
All underlying data have been provided; they are robust, statistically sound, & controlled : OK see below for a few comments
Conclusions are well stated, linked to original research question & limited to supporting results : OK see below for a few comments

Additional comments

Detailed comments:

INTRODUCTION
L 49 Why writing “in hybrid zones” ? The term does not have this meaning in population genetics as hybridogenetic reproduction does not imply classical hybrids or introgression. Indeed you can find “hybrids” (see above for the use of this term in the paper) between e.g. ridibundus and perezi (in your study case) where ridibundus does not occur at all, which is quite wrong for a hybrid zone. Maybe replace by sympatric zones as it is probably more correct to refer to these systems.
LL 57-59 Not that “fixed-in-time” as the parasitic genome is mutating and especially accumulating deleterious mutation as shown by the impossibility of living RR genotypes from GxG crosses which is well explained in your discussion. Please clarify/correct this sentence in the introduction.
LL 60-61 Ref for that? They usually can't colonise much further than their parental species because they need a parental species to parasite (following the current knowledge G can't persist where P is absent).
LL 62-63 Isn’t this sentence a repetition of LL 51-55?
L 64 Replace by “the hybridogenetic reproductive system”
L 73 Replace by “full-klepton population” because such only E populations are not real hybrid populations as real hybrid swarm populations can be.
L91 The native taxa have even been extirpated from part of their distribution by the ridibundus invasion.
L 94-95 Aren’t these systems especially characterized by the absence of introgression despite “hybridization” (see above)?
L 97-98 I can’t get the meaning of this sentence, there is something wrong in the wording and maybe in the placement of it. Maybe replace “already” by “particularly”?
L 103 Please check through the MS that you introduce all abbreviations you use, here “the P-G system” is not introduced above.
L 122-123 Add a mention to the presence of P. saharicus (Doniol-Valcroze et al. 2021, Amphibia-Reptilia) which is concerning due to close phylogenetic (introgression risk) and ecological (replacement risk) affinity with perezi and which could well reach Catalonia from introductions from Northern Africa as in Southern France. Note also that Cuevas et al.’s (2022) method you use would miss the species (perezi-like restriction pattern) if no mtDNA data was added.
METHODS
LL 144-145 Might be useful to explain why using a Saudi Arabian ridibundus sample and what limitation this could cause. Indeed, this sample probably does not belong to the taxa which have been introduced in Western Europe and even not to “real” ridibundus which is a species complex (See Dufresnes et al. 2024, Global Change Biol which could be a reference to add to your paper).
L 185 “for a similar approach” this is quite a weird wording as you basically apply the protocol of this study.
L 188 The diagnosability of kl. hispanicus and moreover “PK” was never tested to my knowledge, at least not in Cuevas et al., and this sentence seems therefore a bit too confident.
LL 199-211 The authors decided to use the SAI-1 length polymorphism to confirm the identification obtained with the first test with RAG1. This cross-checking is very important as Cuevas et al. showed that miss identification can sometimes occur due to restriction failures. Still the authors decided to use SAI-1 even if this method has been regularly shown to be unreliable notably for lessonae and for ridibundus/kurtmuelleri (Cuevas et al. 2022; Dufresnes et al. 2024 Alytes, which needs to be cited here) and avoid using the second step described in Cuevas et al. (2022). Even if I think this does not create any problem with their results the authors should explain their choice and discuss the limitation of the SAI-1 length polymorphism.
LL 213-214 How the samples which were mtDNA barcoded were chosen from the sample pool?
RESULTS
We lack a paragraph or a least a few sentences in part 3.1 about morphological - genetic identification mismatch results (number, any disequilibrium in misidentification, etc.).
LL 259-260 It is interesting to have that much full grafi population which could be studied to know if they are working as “E-systems” in these populations. Following that it could be worth mentioning in the text how many individuals were sampled from these populations in order to have an idea of the potential presence of perezi at low frequency, which is often the case in such systems, and the probability that they had been overlooked/missed during sampling.
L 265 Were these sequences deposited on GenBank? Following Table S1 it seems that one representative of each Haplotype has been published on GenBank, this needs to be explained here. Moreover, I can't find the sequences on Genbank, are they embargoed?
DISCUSSION
LL 291-293 Please develop a little bit more about why Catalonia should be an entry point of the PG-system into Iberia, lower mountains? Crossing of the Pyrenees by water bodies?
LL 306-309 These sentences are a bit messy and could be clarified by clearly saying that morphologically separating R/L/E is easier/more reliable that separating R/P/G. Might be useful to add a sentence about the importance of previous knowledge/training of samplers/identificators, the strict application of published criteria is probably less efficient than years long experience in this kind of tricky species identification.
L 326 Delete “the”
L332 “Val d’Aran” instead of “ Vall d’Aran”
L357-358 Even if this feels quite intuitive a ref should be added here
L368-369 Explain this as it does not seem intuitive
L373-374 And maybe try to understand how 100% grafi populations can persist
L382 citation is Dufresnes et al. 2017 (et al. is lacking)
LL 405-407 This is a very big interpretation which gives a very finalist view of the evolution of kleptons. These meiosis manipulating parasitic hemigenomes are selfish elements and evolve without any “goal” or aim to invade any region. Moreover, as they do not recombine they do not introgress and therefore do not really have a “genetic influence”. I would suggest rewriting this statement to avoid any finalism or to delete this sentence which is a bit of an overstatement.
LL 434-435 Precise the end of the sentence: “[...] do not exist in Catalonia”. Moreover, which evidence do the authors have that their 100% grafi populations are not E-sytem like populations?
L435 Precise the sentence “These results suggest […]”.
L442 Why should this type of mating be unfrequent? Is there any reason to think that one way is preferred to the other (same interrogation at LL 396-397)?
L443 “[...] 22 mitochondrial haplotypes in P. kl. grafi”.
FIGURES
Fig 1. Sea above for the treatment of the “PK system”
Fig 2. Insert map need to be modified. It's impossible to read. Figures 2 and 3 are quite redundants: I would suggest enlarging the taxa distribution map in Fig 2 to make it the only one in the figure and to delete the map of Catalonia as all the information it contains is repeated in Fig 3.
Fig 5. most of the spaces between words are lacking

Reviewer 2 ·

Basic reporting

This article a very good research reference for determining species classification and distribution in the presence of species hybridization. However, there are formatting issues with this article, especially the format of the references, which require further modification.

Experimental design

no comment

Validity of the findings

All comments have already put to the document. Please check it.

Annotated reviews are not available for download in order to protect the identity of reviewers who chose to remain anonymous.

Reviewer 3 ·

Basic reporting

The article entitled Distribution and evolution of the western European water frogs (genus Pelophylax) from Catalonia, northeastern Spain summarizes the distribution, species composition, and genetic diversity of water frog species across the major river basins of the studied region. The paper is well written, clearly structured, and includes a comprehensive review of the relevant available literature. The figures are self-explanatory, of high quality, and well described.
However, one important element seems to be missing: raw morphological data for each individual. The authors compare morphological and genetic species identification, but only a brief table is provided for the morphological identification. An additional table detailing the morphological characteristics of each individual would not only be a valuable resource for the description of Iberian water frogs, but would also allow the authors to further analyze their data and strengthen the impact of the current study.

Experimental design

Similarly, the Methods section appears to lack certain analyses that could strengthen the overall value of the study. Returning to the morphological data, the authors consider morphological identification to be less reliable than molecular methods. However, even in the case of categorical morphological traits, there are analytical approaches—such as Correspondence analysis, with individuals labeled according to genetic assignments—that could be used to quantitatively demonstrate this discrepancy.
In the section addressing mtDNA diversity in P. kl. grafi, basic population genetic metrics, such as haplotype diversity and nucleotide diversity, are missing and should be included to better characterize genetic variation. Moreover, it might be beneficial to compare those metrics with similar metrics of P. perezi or with published values (if any) in other kleptons.
Finally, I would recommend using the length polymorphism of the SAI marker with caution, in light of the findings by Dufresnes et al. (2024, Alytes 41: 5–17).
Minor comment: Please provide the length of the cyt b fragment mentioned in line 213.

Validity of the findings

However, although I consider the methodological approach to be somewhat incomplete, the study presents novel results that are well supported by the data. All research questions were appropriately addressed.

Additional comments

I encourage the authors to consider broadening the analyses performed on their data to further support their conclusions. However, I believe that, after some improvements, the paper will be suitable for publication in PeerJ.

---

## Round 0.2 · Minor Revisions

Thank you for your submission to PeerJ.

I look forward to getting your work back after addressing the comments.

Reviewer 3 ·

Basic reporting

I thank the authors for carefully considering my previous comments. The revised version of the manuscript significantly improves the clarity and overall quality of the text. It is somewhat unfortunate that adequate morphological data could not be collected. I have a few additional comments:

Line 23: I would be careful using klepton PK as a standard klepton in the list, as this observation is based on only two individuals (and several P. r. kurtmuelleri with P. perezi mtDNA). To the best of my knowledge, no other hybrids from this potentially new hybridogenetic system have been recorded so far. Therefore, including klepton PK in the list seems like a strong statement. Also see Dufresnes et al. (2024, Global Change Biology 30, e17180), where only three hybridogenetic complexes are mentioned. Please consider these points and rephrase the text accordingly.

Line 128: I would now consider P. kurtmuelleri a subspecies of P. ridibundus; again, see Dufresnes et al. (2024, Global Change Biology 30, e17180). Please revise the text accordingly.

Line 217: This sentence is a bit confusing. As written, it suggests that P. ridibundus should not be present in the study area, although it was already reported there—as you mention in the introduction—and detecting this species was one of your aims.

Experimental design

-

Validity of the findings

-

---

## Round 0.3 · accepted · Accept

After reviewing this revised version of your manuscript, I see that comments suggested by the reviewers have been included, while the suggestions not considered are justified in detail. Therefore, I am satisfied with the current version and consider it ready for publication.